# Gastric Cancer Vascularization and the Contribution of Reactive Oxygen Species

**DOI:** 10.3390/biom13060886

**Published:** 2023-05-25

**Authors:** Alessio Biagioni, Sara Peri, Giampaolo Versienti, Claudia Fiorillo, Matteo Becatti, Lucia Magnelli, Laura Papucci

**Affiliations:** 1Department of Experimental and Clinical Biomedical Sciences “Mario Serio”, University of Florence, 50134 Florence, Italy; alessio.biagioni@unifi.it (A.B.); giampaolo.versienti@unifi.it (G.V.); claudia.fiorillo@unifi.it (C.F.); matteo.becatti@unifi.it (M.B.); 2Department of Experimental and Clinical Medicine, University of Florence, 50134 Florence, Italy; sara.peri@unifi.it

**Keywords:** gastric cancer, angiogenesis, vasculogenic mimicry, epithelial-to-endothelial transition, hypoxia, reactive oxygen species

## Abstract

Blood vessels are the most important way for cancer cells to survive and diffuse in the body, metastasizing distant organs. During the process of tumor expansion, the neoplastic mass progressively induces modifications in the microenvironment due to its uncontrolled growth, generating a hypoxic and low pH milieu with high fluid pressure and low nutrients concentration. In such a particular condition, reactive oxygen species play a fundamental role, enhancing tumor proliferation and migration, inducing a glycolytic phenotype and promoting angiogenesis. Indeed, to reach new sources of oxygen and metabolites, highly aggressive cancer cells might produce a new abnormal network of vessels independently from endothelial cells, a process called vasculogenic mimicry. Even though many molecular markers and mechanisms, especially in gastric cancer, are still unclear, the formation of such intricate, leaky and abnormal vessel networks is closely associated with patients’ poor prognosis, and therefore finding new pharmaceutical solutions to be applied along with canonical chemotherapies in order to control and normalize the formation of such networks is urgent.

## 1. Introduction

Tumor vascularization plays a fundamental role in cancer progression and metastasis, allowing tumor cells to reach a continuous source of oxygen and nutrients and to evade host immunosurveillance. Sprouting angiogenesis, intussusception, vasculogenesis, vessel co-option, and vasculogenic mimicry (VM) are the main processes that contribute to tumor vascularization, generating an intricate net of vessels, some of which are composed of a mosaic of endothelial and tumor cells. While angiogenesis is commonly described as a complex mechanism including the remodeling of the extracellular matrix, the formation of the Tip-Stalk cells hierarchy and the involvement of pericytes, VM is typically characterized by highly perfused vessels with a significant deposition of matrix proteins, and it is often associated with highly invasive and metastatic tumors, frequently paired with a poor patient prognosis [1]. In previous years, periodic acid-Schiff (PAS)-CD31 was deemed to be the golden standard to distinguish between the two biological processes [2], but recently an alternative method of vascularization, the Epithelial-to-Endothelial Transition (EET), has been observed [3,4]. EET is the acquisition by epithelial cancer cells of endothelial markers, such as CD31, VE-Cadherin, Ephrin A2 and others [3]. Although not completely understood, it has been found that EET is induced by several microenvironmental factors and is established in highly plastic cancer cells, the so-called cancer stem cells (CSCs). One of the master inducers of all the above-described phenomena is hypoxia, which is undoubtedly one of the most typical features of the tumor microenvironment, determining the activation of the hypoxia-inducible factors (HIF) which were first discovered and elucidated by Semenza in 1992, who won the 2019 Nobel Prize in Physiology or Medicine along with Ratcliffe and Kaelin [5,6,7]. In particular, HIF proteins also enhance the stem features of cancer cells and contribute to their differentiation through the EET, by losing cell polarity, increasing the invasive ability and upregulating Twist and Snail, causing the consequent downregulation of the tight junction proteins, E-cadherin and Occludin, together with the upregulation of angiogenesis-related molecules such as VE-cadherin, vitronectin and fibronectin [8]. During severe hypoxic conditions, generated by tumor expansion, the impaired electron transport chain reactions and, more generally, mitochondrial dysfunctions, induce high reactive oxygen species (ROS) production [9] which, consequently, affects cancer cells’ proliferation, migration and metabolism. It is noteworthy that ROS may also be generated during tumor growth by the increased activity of peroxisomes, oxidases, cyclooxygenases, lipoxygenases and thymidine phosphorylase [10]. Gastric cancer (GC), which is currently the fourth leading cause of cancer-related death and the sixth for incidence globally [11,12], is closely dependent on vascularization. Indeed, several studies have showed that *Helicobacter pylori*, which is related to more than half of GC cases, is able to penetrate normal, metaplastic and neoplastic epithelia, triggering an immune-inflammatory response, and thus not only promoting gastric carcinogenesis, but also stimulating the release of cytokines, matrix metalloproteinases and angiogenic factors by gastric epithelial cells upon NF-kB activation [13]. It is well established that *H. pylori*-infected GC patients showed increased tumor vascularization compared to those who underwent *H. pylori* eradication [14]. Moreover, elevated gastrin secretion was also associated with NF-kB activation, as well, enhancing ROS generation and vasculogenesis [15]. When ROS levels were reduced by the use of N-Acetylcysteine, metastatic capability and chemoresistance resulted in inhibition, demonstrating that ROS play a central role in GC pathogenesis. Moreover, it was observed that inoperable GC patients subjected to chemotherapy demonstrated a ‘U-like’ association of mortality rate with vascular density, suggesting that very low and very high vascularization are both linked to poor outcomes [16].

## 2. Tumor Angiogenesis and Vasculogenesis

Angiogenesis is a multistep process that involves the formation of new blood vessels starting from pre-existing ones. During cancer development, the tumor mass constantly grows until the oxygen concentration is too low and the catabolic products of the accelerated glycolysis affect the microenvironmental pH. In particular, hypoxia often triggers the early endothelial response by activating the production of pro-angiogenic factors such as the vascular endothelial growth factor (VEGF), angiopoietin-2 (ANG-2), or fibroblast growth factor-2 (FGF-2) [17]. These signals wake up the pericytes, detaching them from the vessel wall and allowing the endothelial cells to degrade the basement membrane through the activation of matrix metalloproteinases (MMPs). After such events, the endothelial cells lose their junctions, the vessel dilates and several plasma proteins extravasate, creating a new temporary matrix layer for ECs migration [17]. The endothelial cell closest to the higher VEGF concentration gradient is then selected by the Notch-Dll4 axis as the “Tip” cell [18], which will lead to the building of the nascent vessel, while neighbor cells will be consequently selected as “Stalk” cells and they will divide to follow the Tip. Once the vessel is mature, all the junctions between ECs have been restored, pericytes have covered the vessel again and the basement membrane has been deposited, the blood will flow and the new vessel will be completely perfused. However, while in physiological conditions Dll4 selects the Tip cell only, activating the Notch pathway, in tumor-induced angiogenesis Dll4 expression was observed in the majority of tumor vessels [19]. Sprouting angiogenesis is a complex and long process, especially in tumors, where ECs often compete for the leading positions, generating a continuous change between Tip and Stalk status. Therefore, endothelial tip and stalk cells’ fate determination is not fixed by the initial conditions, but rather there is a constant dynamic phenotype switch [20]. Despite some interesting clinical trials aimed at treating and controlling angiogenesis in GC, their efficacy in improving patients’ Overall Survival (OS) remains limited. In 2011 and 2014, the trials AVAGAST and AVATAR [21,22] demonstrated that the use of Bevacizumab, a monoclonal antibody inhibiting VEGF-mediated angiogenesis by binding and inactivating the VEGF-A ligand, was able to modestly increase the Overall Response Rate (ORR) and the Progression-Free Survival (PFS) but with no significant improvement in the OS. Moreover, it clearly appeared that the efficacy was dissimilar depending on territorial differences, which might be due to different hospitalization conditions or genetic discrepancies [23]. Better results were achieved through the trials REGARD and RAINBOW [24,25], which tested Ramucirumab, a human monoclonal antibody binding to the extracellular region of VEGFR-2 and blocking the downstream effects of VEGF, improving PFS and OS, especially when combined with Paclitaxel. Such a promising result induced the American Society of Clinical Oncology to confirm Ramucirumab as the second-most-effective targeted drug after Trastuzumab [26]. Although tyrosine kinase inhibitors (TKi) are not widely used in the treatment of GC, here we report Sunitinib, one of the first TKi approved for use in imatinib-resistant GISTs. Indeed, it not only blocks VEGFRs, PDGFR-*α*, PDGFR-*β* and c-Kit [27] molecules involved in the vasculogenic process, but also exerts a possible ROS-mediated cytotoxic effect, albeit one that is still not completely clear [28]. We also need to mention that such anti-angiogenic therapies often result in only a transitory improvement of the clinical picture. Resistance to such regimens is divided into two main categories: the first is defined as the evasive or adaptive resistance to angiogenesis inhibitors, which involves revascularization due to the upregulation of alternative pro-angiogenic signals, the protection of the tumor vasculature by recruiting pro-angiogenic inflammatory cells or by increasing protective pericyte coverage, the increased invasiveness of tumor cells into adjacent tissues to co-opt normal vasculature and the augmented metastatic relapse and tumor cell growth in lymph nodes and distant organs. The second one is called intrinsic resistance, and includes those individuals who had never benefited from the anti-angiogenic treatments due to an innate indifference, probably due to the redundancy of several pro-angiogenic signals, an inflammatory cell-mediated vascular protection, and the invasive angiogenic-independent co-option of normal vessels [29].

## 3. Vasculogenic Mimicry

Described for the first time in uveal melanoma by Maniotis et al., VM is a biological mechanism exploited by several cancer histotypes to generate PAS-positive vessels lined by tumor cells [30]. Although early evidence indicated that the inner wall of such vessels was composed of only tumor cells, Chang et al. demonstrated that the luminal surface might indeed be lined by an intricate mosaic of tumor and endothelial cells [31]. The intricate transformation that leads cancer cells to perform VM is, to date, not fully understood, but accumulating evidence points to a close correlation between CSCs and VM formation. Indeed, during VM, high plasticity cancer cells that do not express any typical endothelial markers behave like proper endothelial cells to achieve new sources of oxygen and nutrients. To distinguish between the two histotypes, the golden standard is the immunohistochemical analysis of two endothelium-related proteins (CD31/CD34) coupled with the PAS reaction [2]. Currently, very little is known about the formation of VM in GC. A close association has been reported between GC patients with VM and the formation of hematogenous metastasis, probably due to the aggressive features developed by cancer cells and to the fact that, when lining the luminal surface of the vessels, cancer cells are directly exposed to the bloodstream, increasing their proficiency to detach and form distant metastases [32]. MMP-2 and MMP-9 have been positively correlated with VM, conferring on cancer cells the capability to remodel the extracellular matrix and degrade the vascular basement membrane [33], while EphA2 has been reported to be directly involved in the formation of tubular networks [34]. These phenomena were confirmed by Sun et al., who demonstrated that out of a collection of 84 specimens of gastrointestinal stromal tumors (GIST), 21 were found to be VM positive, with a high production of MMP-2 and MMP-9 [35]. Although GISTs are typically not aggressive tumors, patients with VM-enriched vessels experienced a worse prognosis with respect to the ones with low VM-dependent vascularization. VM is commonly associated with advanced-stage GCs and with patients’ poor prognosis, although the vast majority of the data are available only in Eastern countries [36]. Moreover, one of the main issues in identifying VM in GC samples is the heterogeneous mucous tissues of the gastrointestinal tract, which might lead to false PAS-positive results, thus overestimating the VM histological grade. It is today well-known that cells resistant to anti-angiogenic therapies, such as those reported above, are often more prone to generate VM networks in vivo and in vitro, and therefore research into new therapeutic strategies to target both phenomena at the same time is urgently needed.

## 4. Epithelial-to-Endothelial Transition

Under particular microenvironmental conditions, i.e., hypoxia, high tumor interstitial fluid pressure and altered extracellular matrix, VM-prone cells begin to express typical endothelial antigens through the so-called Epithelial-to-Endothelial transition. Hypoxia is the best-known and most potent VM inducer, activating HIF-1α and HIF-2α, which in turn bind to the hypoxia-response elements of target genes such as VEGF, VEGF receptors, EMT inducers and stem-associated genes [37]. Under hypoxic conditions, Twist1 translocates into the nucleus and promotes VE-Cadherin expression, triggering the transition of epithelial cells to an endothelial-like phenotype [38,39]. Indeed, VE-Cadherin activates PI3K via EphA2 phosphorylation, promoting the matrix metalloproteinases secretion and thus extracellular matrix remodeling [40]. In particular, HIF-1α involvement was reported to promote the stabilization of Notch, a typical endothelial antigen, by binding to its intracellular domain and consequently promoting Nodal transcription [41]. It was recently reported that the onset of chemoresistance, and in particular to 5-Fluorouracil, might induce the EET in AGS cells via the upregulation of TYMP, EphA2 and VEGFR2 [42]. However, the EET is still a poorly understood phenomenon, with narrow limits that denote the difference with a proper VM process. Additionally, from the clinical point of view, the treatment of such a biological event is currently debated as all the above-described anti-angiogenic therapies might induce the onset of VM cells due to the generation of hypoxic regions. Even after discontinuing treatments, endothelial vessels might rebound and link to the neo-formed VM channels [39]. To date, the only therapeutic options evaluated in vivo are the use of Doxycycline, a tetracycline derivative, acting by the inhibition of the degradation of E-Cadherin preventing in this way both the EET and VM [43], anti-Notch4 antibodies which downregulate Nodal expression [41] and dual antiplatelet therapy, a gamma-secretase inhibitor, which was found to inhibit glioblastoma CSCs from differentiating into endothelial-like progenitor cells through blockade of Dll4-Notch signaling [44]. However, no therapies nor experimental drugs are currently reported for GC treatment.

## 5. Innovative Pharmacological Approaches

Even though no clinical trials with anti-VM drugs have been reported, we describe here the use of Rapamycin, Genistein and Thalidomide. Rapamycin, a macrolide compound used to coat coronary stents and to prevent organ transplant rejection, has a strong immunosuppressive effect and acts by targeting mTOR, which is a crucial VM and angiogenic effector, especially when induced by hypoxic conditions [45,46]. Similarly to Rapamycin, Everolimus, and Temsirolimus act as mTOR inhibitors for GIST treatment, although it has still not been clarified whether their antitumor effect might be produced by ROS mediation [47]. The use of EF24, a new curcumin analog, was evaluated by Chen et al. in combination with Rapamycin for the treatment of advanced GC. Indeed, they demonstrated that EF24 acted as an ROS inducer, sensitizing GC cells to Rapamycin-induced growth inhibition, decreasing the mitochondrial membrane potential and leading cancer cells to apoptosis [48]. The co-treatment of GC cells with MK-2206, an Akt inhibitor, and EF24 induces ROS generation via the stimulation of the endoplasmic reticulum through the activation of CHOP and ATF-4, thus decreasing the mitochondrial membrane potential and inducing apoptosis via the Bcl-2/Bax protein ratio dysregulation [49]. Recently, another curcuminoid analog, the B63 targeting TrxR1, has been reported to induce ROS-mediated paraptosis-like cell death in GC cells, leading to the reduction of tumor growth in vivo [50]. Genistein is a phytoestrogenic compound derived from soybean and is already used in many in vitro experimentations in GC to induce apoptosis in a dose and time-dependent manner by downregulating the expression of the antiapoptotic Bcl-2 protein and upregulating the expression of proapoptotic Bax [51]. Physiologically, Genistein is mainly metabolized through oxidation, sulfation, glucuronidation, hydroxylation or methylation, and therefore its derivatives’ effects are still not well evaluated (although it was speculated that the 5,7,39,49-tetrahydroxyisoflavone (THIF) and 2 glutathinyl conjugates of THIF might be responsible for the angiogenesis inhibition and reduction in endothelial cell proliferation) [52]. Moreover, it has been shown that THIF may induce the cycle arrest, activate the p38 MAPK signaling pathway and induce DNA strand breakage via ROS formation [53]. Moreover, increasing evidence suggests that gastric CSCs may be particularly sensitive to a low dose of genistein (15 mM), leading to inhibited self-renewal ability, drug resistance and tumorigenicity due to the suppression of ABCG2 expression and ERK1/2 activity [54], and to the downregulation of Gli1 and CD44 paired with a reduced migration capability [55]. Thalidomide, involved in the 1950 scandal of several birth defects, is still used for the treatment of some multiple myeloma cases even though its pharmacologic action and targets are not completely clear. Kita et al. reported that several N(α)-phthalimide glutarimide derivatives, similar to Thalidomide, are able to selectively target the platelet-derived endothelial cell growth factor (PD–ECGF), which is structurally identical to the thymidine phosphorylase (TYMP), an important enzyme for purine synthesis and one closely involved in angiogenesis response [56]. It was also demonstrated that Thalidomide in GC cells might partially revert chemoresistance to 5-Fluorouracil and inhibit invasion, migration and VM capabilities [42]. All the above-described anti-angiogenesis and anti-VM therapies are summarized in Table 1.

## 6. Gastric MALT Lymphoma

A particular and noteworthy case is gastrointestinal lymphomas (indolent B-cell non-Hodgkin), of which the so-called MALT lymphoma (mucosa-associated-lymphoid tissue) is the most common type. Such a kind of lymphoma may arise from any mucosal tissue, but often is associated with the gastric one, where memory B cells may be subjected to chronic inflammation and infection in the extranodal marginal zone of the mucosa-associated lymphoid tissue [57]. Etiological agents such as the *H. pylori* may play an important role in the pathogenesis of this disease. Indeed, *H. pylori* exposition leads to an increased MALT accumulation and to the stimulation of tumor-infiltrating T cells, which generate the chronically inflamed microenvironment [58]. The continuous antigenic stimulation, which promotes B cell proliferation and neutrophils retrieval, induces the B cell population to develop oligoclonal and/or monoclonal sub-populations, leading to the oncogenic event [59]. Moreover, the release of ROS by neutrophils might spread a wide genotoxic effect, allowing the amplification of genetic aberrations [60]. Currently, the gold standard therapy for *H. pylori*-positive gastric MALT lymphoma is based on a combination of three drugs, a proton pump inhibitor, clarithromycin, and amoxicillin or metronidazole, while in the case of failure or for *H. pylori*-negative MALT, patients commonly undergo involved-site radiation therapy (ISRT) or Rituximab [61]. The MALT being histologically characterized by an abundant distribution of the microvascular network, comprising immature capillaries, lymphatics, and venules, Nakamura M et al. proposed a new pharmaceutical approach based on an anti-VEGF antibody, such as bevacizumab or aflibercept, and celecoxib, a cyclooxygenase-2 inhibitor with promising results [62].

## 7. Gastrointestinal Stromal Tumors

Gastrointestinal stromal tumors (GIST) are the most common mesenchymal cancers of the gastrointestinal tract, albeit accounting for less than 1% of all gastrointestinal tumors [63]. They are classified as soft tissue sarcomas, and frequently arise from Cajal cells along the digestive tube or, less frequently, from telocytes or smooth muscle cells [64,65,66]. Over the years, GIST have been more and more precisely characterized from a molecular point of view, and thus now they are enclosed in two main categories depending on the proto-oncogene (KIT) or the platelet-derived growth factor receptor alpha gene (KIT/PDGFRA) mutations. The ones (5–7.5% of all GIST) with KIT/PDGFRA wild-types are then sub-classified depending on the functionality of the succinate dehydrogenase enzyme (SDH B/C/D), which might be deficient due to mutation or epigenetic regulation. In the rare cases when no KIT/PDGFRA mutations are detected and the SDH complex is functional, several very rare driver alterations could have occurred in genes such as the RAS gene family, BRAF, NF1, NTRK1–3, and FGFR1–4 [67]. To date, surgery is the only therapeutic option in combination with adjuvant chemotherapy based on the tyrosine kinase inhibitor (TKi) Imatinib mesylate for KIT/PDGFRA-mutated patients [68]. Xu et al. recently revealed that the chronic administration of Imatinib might induce ROS accumulation over time, leading to an increase and stabilization of HIF-1α, which in turn stimulates the phosphogluconate dehydrogenase (PGD), resulting in a metabolic switch from a typical tricarboxylic acid cycle with high oxidative phosphorylation to the exploitation of the pentose phosphate pathway. PGD was observed to be upregulated in GIST, and especially in GIST cell lines resistant to Imatinib when compared with sensitive ones, promoting cell proliferation and apoptosis avoidance [69]. Indeed, a higher PGD expression produces an abundant quantity of NADPH, providing reducing intermediates for the glutathione and thioredoxin systems and, at last, reducing the ROS-dependent apoptosis and cell cycle arrest. Angiogenesis is another important force affecting GIST malignancy [70]. For example, it was reported that BRD4 upregulation enhanced GIST migration and invasion by regulating angiogenesis through the NF-kB/CCL2 signaling pathway, while CCL2 is also useful to recruit tumor-associated macrophages, increasing the cancer microvessel density and secreting several pro-angiogenic molecules such as VEGFA, LOX and MMP9 [32,71]. Moreover, mutations in the PPP2R1A gene are associated with a more aggressive tumor phenotype causing an increased growth rate via stimulating the phosphorylation of c-kit, Akt1/2, ERK1/2 and WNK1, of which the latter one regulates the angiogenetic response [72,73]. Furthermore, GIST has also demonstrated a high capability for assessing vasculogenic mimicry, as demonstrated by Sun B et al., observing that, in a retrospective study of 84 patients with GIST, VM is associated with tumor size, mitotic rate and liver metastasis [35]. VM-positive GIST patients correlate with a high risk of metastatic progression and with poor prognosis according to the survival Kaplan–Meier curves. They also found an upregulation and activation of MMP-2 and MMP-9, whilst not detecting any significant change in VEGF expression. It was recently observed that epigenetic regulation might also play a major role in GIST vascularization control, as the upregulation of the histone demethylase KDM4D was demonstrated to affect promoters of the H3K9me3 and H3K36me3 genes, enhancing the angiogenesis in vivo through the HIF1β/VEGFA pathway stimulation, leading to the overexpression of CD31 [74]. It is also noteworthy that several TKi exert an anti-angiogenic effect, too. Indeed, Cabozantinib, as well as sorafenib, targets VEGFR2, Flt-3 and c-Kit, leading to a diminished tumor microvascular density [75,76].

## 8. ROS-Induced Vascularization

Hypoxia in varying degrees commonly generates Reactive Oxygen Species (ROS), probably affecting the complexes I, II and III of the mitochondrial electron transport chain (ETC), even though the mechanism is still debated [77]. ROS are molecular oxygen-derived molecules that are formed by reduction-oxidation reactions or by electronic excitation. They contain one or more unpaired electrons, and are characterized by a highly reactive status [78]. Cellular endogenous ROS are generated physiologically in the process of mitochondrial oxidative phosphorylation [79], as well as in cellular responses to xenobiotics, cytokines, viruses and bacterial invasion, or they may arise from interactions with exogenous sources [80]. When ROS concentration overwhelms the cellular antioxidant defense system, oxidative stress occurs, inducing damage to proteins, lipids and nucleic acids, contributing in this way to the pathogenesis of several human diseases such as neurodegenerative [81,82], cardiovascular [83,84], and inflammatory diseases [85], infertility [86,87], aging [88,89] and cancer [90]. Three main classes of ROS are commonly produced: hydroxyl radicals characterized by a high reactivity and short half-life, superoxides with low reactivity and the capacity to diffuse through anionic channels and hydroperoxides with moderate reactivity and the capability of diffusing via aquaporins [9]. Superoxide, hydrogen peroxide and peroxyl induce membrane lipid peroxidation, generating unstable fatty acid radicals (lipid hydroperoxides) which rapidly degrade into cytotoxic ketones, epoxides and reactive aldehydes released in the gastrointestinal compartment [91]. Their plasma and urinary levels are indeed elevated due to the absorption in the intestines, leading to a higher risk of mutagenesis and cancer progression [92,93]. Moreover, ROS’ intracellular accumulation results in the stabilization and activation of HIF-1α and the degradation of SIRT3, leading to the transcription of VEGF, LDHA and PDK1, even under normoxic conditions, promoting in such a way tumor angiogenesis and progression in GC [94]. In particular, HIF-1α and HIF-2α regulate tumor cell proliferation, migration, glycolysis and angiogenesis via endogenous ROS production [95]. Moreover, Park et al. demonstrated that endogenously produced ROS or stimulation by *H. pylori* stabilize HIF-1α protein in human GC cells even under normoxic conditions via non-mitochondrial ROS [96]. Indeed, *H. pylori* is capable of producing ROS and reactive nitrogen species (RNS) or inducing their production in neutrophils, endothelial cells, and gastric mucosal cells [97]. Such particular bacteria are capable of survive in the host stomach, inducing the expression of inducible NO synthase (iNOS) in gastric mucosal epithelial cells, vascular endothelial cells, or infiltrated inflammatory cells [98]. *H. pylori* then translocates into the cells through the cytotoxin-associated gene A (CagA), inducing the production of ROS and triggering, in turn, a cascade response including the stimulation of cell cycle progression, and therefore proliferation [99], and damages to mitochondrial and nuclear DNA [100]. Cation transport regulator 1 overexpression in AGS cell lines infected with CagA-positive *H. pylori* has also been demonstrated to reduce cellular glutathione levels via the glutamylcyclotransferase, leading to the rapid accumulation of ROS [101]. Such damages, which rapidly induce mutations and carcinogenesis, are also generated through ROS promotion by another important *H. pylori*-associated virulence factor [102], the vacuolating cytotoxin A (VacA) [103]. The combined effect of CagA and VacA is thus reported to decrease cell autophagy in gastric epithelial cells by reducing intracellular glutathione levels, increasing ROS accumulation, enhancing the resilience of *H. pylori* infection in the stomach, and therefore promoting GC development [104]. On the other hand, VacA inhibits the expression of integrin-linked kinase and endothelial nitric oxide synthase, leading to decreased ROS production in macrophage/monocyte lineages in order to evade immune system surveillance [105]. Recently, proton pump inhibitors have been used for the treatment of peptic ulcers and for the *H. pylori* eradication therapy, being capable of suppressing the effect of the bacterial-induced acid secretion. Rabeprazole has been reported to reduce *H. pylori*-induced gastric mucosal damage by increasing the concentration of reduced glutathione [106], while lansoprazole and omeprazole have been observed to inhibit inflammatory cytokines’ production by stimulating gastric epithelial cells and endothelial cells via the blockage of redox-sensitive transcription factor activation [107]. Lian et al. demonstrated the complex interplay between ROS and angiogenesis in GC, exposing the AGS cell line to nicotine and treating endothelial cells with the resulting tumor cell-derived conditioned media. Indeed, they observed that nicotine promoted endothelial cells’ growth and tube-like formation, induced by IL-8 expression via the stimulation of ROS/NF-κB and ROS/MAPK (ERK1/2, p38)/AP-1 axes [108]. Gastrin, a major component of the gastric juice, promotes angiogenesis by activating HIF-1α/β-catenin/VEGF signaling in GC [109] and metastasis via the β-catenin-TCF4 pathway [110]. Liu et al. demonstrated that gastrin regulates IκBα and NF-κB in GC cells, enhancing ROS generation and controlling the anti-apoptotic Bcl-2 and pro-apoptotic Bax expression in a ROS-dependent manner [15]. It has also been reported that EMT and metabolism alterations may depend on the ROS/NF-κB/HIF-1α axis. Indeed, Qin et al. demonstrated that N-acetylcysteine-dependent ROS scavenging diminished NF-κB and HIF-1α activation in autophagy-deficient GC cells, preventing the metabolic switch (from mitochondrial oxidative phosphorylation to aerobic glycolysis), which is closely associated with malignity and chemoresistance in GC [111]. Radiotherapy, as well as platinum-based drugs, might induce ROS generation by high energy ionizing radiation [112] or via NADPH oxidase, respectively [113,114]. In this context, it has been already demonstrated that Oleocanthal, a minor polar compound extracted by extra-virgin olive, causes cell cycle inhibition and ROS accumulation in AGS cells resistant to 5-Fluorouracil and Paclitaxel, but not to Cisplatin. Indeed, Cisplatin-resistant cells seem to be characterized by higher levels of antioxidant enzymes that are capable of counteracting the Oleocanthal-induced ROS accumulation [115]. Endogenous ROS, which are able to directly affect GC cells, can also modulate tumor microenvironmental components. Indeed, it was observed that tumor-infiltrating lymphocytes might be attracted by ROS, exerting their typical antitumor effect [116]. Radiotherapy can damage cancer cells’ DNA, dramatically increasing ROS levels, inducing vascular trauma, tissue self-healing, edema and immune cell infiltration, and generating, in turn, an increased demand for oxygen with consequent HIFs activation [117]. The immune cell infiltration, commonly composed of tumor-associated macrophages, T-cells, B-cells, and myeloid-derived suppressor cells, is influenced and recruited by severe hypoxic conditions [118]. The interaction among ROS, angiogenesis and inflammation is an important pathogenic factor for GC progression, as inflammatory mediators can regulate ROS/RNS production [119]. ROS-activated TNF-α has been reported to downregulate IκBα, mediating, in turn, the release of inflammatory mediators such as NOX2, IL-6, IL-2, IL-8 and CXCL12 [120]. Several enzymes contribute to their production in three main cellular districts (mitochondria, cytosol and single membrane-bound organelles (peroxisomes, endosomes and phagosomes)), and thus ROS may diffuse through specific channels or be delivered by exosomes [121]. ROS play also a major role in tumor angiogenesis by controlling VEGF expression. As reported by Xia et al. the NADPH oxidase inhibitor DPI and the complex I inhibitor Rotenone are capable of inhibiting VEGF protein and mRNA levels, probably via HIF-1α, reducing neovascularization and tumor growth [122].

## 9. ROS Scavenging Activity

There are three main enzymatic components of the antioxidant defense system: the metal ion-dependent superoxide dismutases (SODs) [123], the catalase and the glutathione peroxidase (GPx) [124]. To the first family belongs two copper/zinc-containing members: CuZnSOD (SOD1), localized within the cytosol, mitochondrial intermembrane space and nucleus, and the extracellular SOD (EcSOD or SOD3), which is the predominant antioxidant enzyme secreted into the extracellular space and the manganese-containing MnSOD (SOD2), active into the mitochondrial matrix. Such particular enzymes require specific metal cofactors for catalyzing the redox process of dismutation, i.e., the conversion of superoxide molecules into oxygen and hydrogen peroxide. Of particular interest is that EcSOD is commonly reported to be down-regulated in almost all cancer histotypes, albeit increased in gastric adenocarcinoma and prolactinoma, a benign pituitary gland tumor [125,126]. However, the cause of such an increase in serum EcSOD remains unclear. Another fundamental enzyme with scavenging activity against intracellular ROS is the catalase, localized in the peroxisomes, which converts hydrogen peroxide into water and oxygen. While SOD and catalase do not need co-factors to function, GPx requires the co-factors and proteins belonging to the glutathione system (glutathione reductase and glucose-6-phosphate dehydrogenase) to convert the hydrogen peroxide in water [127]. Even though the importance of such a scavenging system is clearly remarkable, to date no scientific articles have been produced about gastric cancer, and here lies the urgent need to stimulate the worldwide community to improve the research on such an interesting topic.

## 10. The Role of NADPH Oxidase in GC

NADPH oxidases, one of the major ROS sources, are multi-subunit enzyme complexes catalyzing the production of a superoxide-free radical by transferring one electron to oxygen from NADPH with the extrusion of an H^+^ molecule. This highly conserved protein family includes seven oxidases, namely NADPH oxidase 1 (NOX1), NOX2, NOX3, NOX4, NOX5, Dual oxidase 1 (DUOX1) and DUOX2. It has been reported that p22phox expression, an essential component of the membrane-associated enzyme NADPH-oxidase NOX1, NOX2 and NOX4, is more highly expressed in actively proliferating endothelial cells than in quiescent cells, demonstrating that ROS produced by NADPH oxidase might sustain endothelial cells’ proliferation [128]. NOX1 plays an important role in endothelial cell proliferation, sprouting and migration by increasing ROS-sensitive transcription factors and ROS-dependent phosphatases that block PPAR-α activity, which in turn is responsible for NF-kB inhibition [129]. On the other hand, NOX2 is a critical ROS source in endothelial cells and an important regulator of their function, as its genetic deficiency, i.e., the loss of gp91phox expression, leads to the chronic granulomatous disease causing enhanced endothelium-dependent vasorelaxation, the reduction of vascular aging markers and oxidative stress by limiting NO bioavailability [130], whilst NOX5 is actively involved in ROS-mediated proliferation and the formation of capillary-like structures in human microvascular endothelial cells [131]. NOX3, DUOX1 and DUOX2’s involvement in cancer angiogenesis has not yet been fully established (Figure 1). You et al. demonstrated that NOX1/2/4 mRNA expression levels in GC tissues were higher than in normal tissues, while NOX5 and DUOX1/2 expression levels were lower. Moreover, they observed that a high NOX2 mRNA expression was associated with better OS, whilst NOX4 and DUOX1 were closely correlated with a worse outcome, in particular in intestinal-type GC patients [132]. Similarly, Qiao and colleagues investigated the role of NOX4 in GC by observing its upregulation in the tumor mass compared with adjacent non-tumor tissues, which was correlated with higher invasive capability and a worse TNM stage [133]. Indeed, studies indicated that lipopolysaccharides from *H. pylori* induced NOX1-derived ROS through TLR4 in guinea pig gastric pit cells, in this way exploiting NOX1 as the trigger of the innate immune responses against *H. pylori* [134]. Moreover, high levels of NOX1 were detectable in GC cells (intestinal, diffuse or signet-ring cell type) but not in normal gastric mucosa cells [135]. This points to an intriguing possibility that GC undergoes aberrant control of NOX1 expression. Even though all these proofs clearly determine the deep involvement of the NOX family, and in particular NOX1, in ROS-mediated GC progression, few works in the literature are available to date. Of particular interest is the biological process termed “ROS-induced ROS release” (RIRR), in which one cellular compartment generates and releases ROS, stimulating ROS production by another adjacent compartment or organelle. This phenomenon was initially described in cardiomyocytes, demonstrating that upon the mitochondrial permeability transition pore opening the mitochondrial membrane potential rapidly dissipates, triggering the so-called “burst phase” of ROS generation, which caused a synchronous phenomenon in the contiguous mitochondria [136]. Indeed, this event might be considered a natural safety valve that prevents excessive ROS accumulation inside the mitochondria, through the occasional opening of their pores to release ROS. During pathologic conditions, i.e., cancer, such a release might trigger irreversible positive feedback, leading to the activation of apoptotic or autophagic pathways [137]. Moreover, Young-Mee et al. reported that, in endothelial cells, NOX2 is capable of sensing NOX4-derived H2O2, inducing mitochondrial ROS production via pSer36-p66Shc and, lastly, enhancing VEGFR2 activation, thus amplifying the angiogenesis signaling program [138]. Amongst the currently available clinical approaches, we report the use of Apatinib, which is thought to be one of the most promising drugs, being a well-known anti-angiogenic agent as a VEGFR2 selective inhibitor. Already approved for the treatment of metastatic GC by the FDA [139], it is reported to affect ROS production via GPX4 and SREBP-1a [140].

## 11. Conclusions

To date, the relationship between ROS and tumor vascularization is still debated, but the urgency of learning how to treat and control cancer-induced angiogenesis, especially when particular hostile conditions, such as hypoxia, start to be established in the tumor microenvironment, is clear. Several attempts have been made to counteract tumor angiogenesis, for example through treatment with Bevacizumab and Ramucirumab, which have proven to be effective in the early tumor stages, but which often induce hypoxic regions, selecting, in such a way, a pool of cancer cells with an endothelial behavior that in turn might replace the proper endothelium. VM and EET molecular processes also need to be further clarified in order to fully understand their genesis and the real contribution of external factors and stimuli which might induce each kind of phenotype.

## Figures and Tables

**Figure 1 biomolecules-13-00886-f001:**
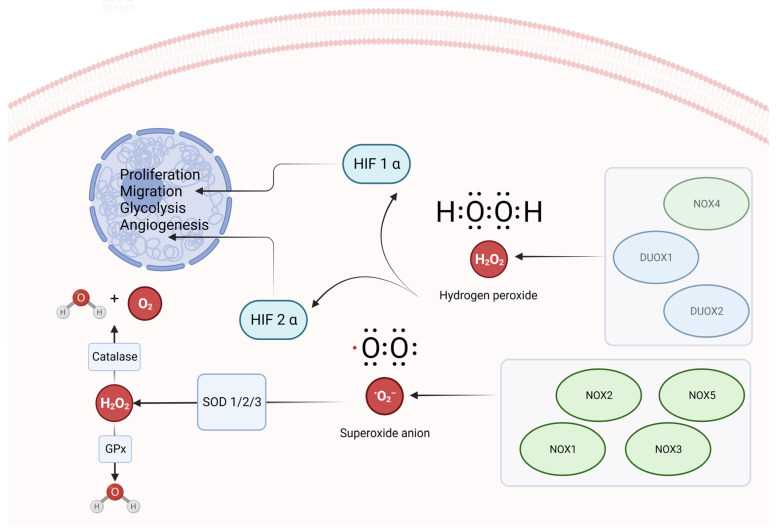
A schematic representation of the NOX and DUOX-mediated ROS generation, their interaction with HIF-1α and HIF-2α, and the ROS scavenging system in GC cells.

**Table 1 biomolecules-13-00886-t001:** Chemotherapies aimed to control tumor-induced vascularization.

**Treatment**	**Target**	**Inhibited Phenomenon**	**Reference**
Bevacizumab	VEGFA	Angiogenesis	[21,22]
Ramucirumab	VEGFR2	Angiogenesis	[24,25]
Rapamycin	mTOR	Angiogenesis/VM	[45,46]
Genistein	Bcl-2/ABCG2/ERK1/2/Gli1/CD44	Angiogenesis/VM	[51,52,54,55]
Thalidomide	Unknown	Angiogenesis/VM	[43,57]
Doxycycline	E-Cadherin	Angiogenesis/VM/EET	[41]
Anti-Notch4 Antibodies	Notch4	Angiogenesis/EET	[43]
Dual Antiplatelet Therapy	Dll4-Notch axis	Angiogenesis/EET	[44]

## Data Availability

Not applicable.

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
