# Peer review of "Gastric Cancer Vascularization and the Contribution of Reactive Oxygen Species"

_biomolecules, 2023, doi:10.3390/biom13060886_

Round 1

Reviewer 1 Report

The subject of this review is a very interesting one. There are problems however in the way the paper has been written.

Approximately two thirds of the text are on the vascularization of tumours in general, rather than on the subject of the paper. There are already many reviews on the subjects and to repeat in this paper does not add anything. Furthermore, this section overlooks completely the role of non-angiogenic tumours and vascular co-option, which is by far more widespread of the vascular mimicry. Therefore, the description they show of the vascularization of tumours is incorrect (see detailed notes below).

I would advise the authors to scrap the first two thirds and to concentrate on the actual section on ROS in gastric cancer which is instead a less known ad very interesting topic. There are also relevant papers on the topic not cited and would be interesting to include them. 

I would like to propose an outline of the paper as follow (just as a suggestion):

Introduction (brief)

-Role of ROS in cancer and effect on vascularization

Vascularization of gastric  tumours: what do we know

-I would review carcinoma, MALT lymphoma and GIST

ROS production and Helicobacter Pylori

This is a very interesting point

ROS in gastric tumours

-Role in vascularization and oncogenesis

The Role of NADPH Oxidase in GC

Possible new development for treatment

Conclusions

Specific comments:

Abstract

Lines 18-20:  wrong definition of vasculogenic mimicry. The correct definition is given later on (lines 125-127).

1 Introduction

Lines 31-33. The authors have forgotten non-angiogenic growth by vessel co-option!!

2 Tumour angiogenesis and vasculogenesis

Lines 86-88: This is by now incorrect. Hypoxia is present also in non-angiogenic tumours.

Lines 105-123: Several mechanisms of resistance to anti angiogenic treatment have been now described. The authors should quote and briefly discuss them

3 Vasculogenic mimicry

Lines 128-129: Mosaic lining of the vasculogenic mimicry channels happens but not always!!

Lines 153-156: It is correct that it is a mechanism of resistance to anti angiogenic treatment but not the only or  commonest (see comment above).

Lines 156-195. This sections perhaps should be self-contained, as possible new approaches to gastric cancer. It begins with drugs which could possibly interfere with VM and than move on to other drugs, to end with anti angiogenic approaches. It should not be in the VM paragraph.

4 Epithelial to Endothelial transition

Lines 199-200: “Hypoxia is the best known and most potent VM inducer,” : reference 52 does not discuss vasculogenic mimcry.

Lines 215-216: This event can happen, but not always.

5 ROS induced vascularization.

Line 229-231: Again, the non angiogenic growth by vessel co-option is omitted, making this sentence rather misleading.

6. The Role of NADPH Oxidase in  GC

Lines 358-359 “ Finally, ROS-induced angiogenesis is still a debated topic and therefore few drugs are currently investigated to inhibit such a phenomenon. “

This statement is puzzling as already in the abstract the authors stated:

“ In  such a particular condition, reactive oxygen species play a fundamental role, enhancing tumour proliferation, migration, inducing a glycolytic phenotype and promoting angiogenesis.”

Author Response

Approximately two thirds of the text are on the vascularization of tumours in general, rather than on the subject of the paper. There are already many reviews on the subjects and to repeat in this paper does not add anything. Furthermore, this section overlooks completely the role of non-angiogenic tumours and vascular co-option, which is by far more widespread of the vascular mimicry. Therefore, the description they show of the vascularization of tumours is incorrect (see detailed notes below).

We thank the reviewer for the comments and suggestions aimed to improve our review article. We acknowledge that the first part of our article is focused on tumor vascularization and the relative clinical therapies but we think that it is mandatory, albeit redundant with other reviews, to introduce “step-by-step” all the most known angiogenic processes and which are the drugs currently available to counteract each phenomenon. Therefore, to be more focused on the proposed topic, we changed the title, which might be misleading, to “Gastric Cancer Vascularization and the Contribution of the Reactive Oxygen Species”. Moreover, we added the vascular co-option phenomenon.

I would advise the authors to scrap the first two thirds and to concentrate on the actual section on ROS in gastric cancer which is instead a less known ad very interesting topic. There are also relevant papers on the topic not cited and would be interesting to include them.

I would like to propose an outline of the paper as follow (just as a suggestion):

Introduction (brief)

-Role of ROS in cancer and effect on vascularization

Vascularization of gastric  tumours: what do we know

-I would review carcinoma, MALT lymphoma and GIST

ROS production and Helicobacter Pylori

This is a very interesting point

ROS in gastric tumours

-Role in vascularization and oncogenesis

The Role of NADPH Oxidase in GC

Possible new development for treatment

Conclusions

We thank the reviewer for the precious comment. Albeit the structure of the article provided by the reviewer is very interesting and punctual, we need also to take into consideration both the revisions provided by the reviewers. Thus, as the second reviewer seems satisfied with the organization of our article we tried to satisfy the request of the first reviewer by adding the suggested topics. Therefore, following the reviewer's suggestion, we added several new topics and points of discussion. By adding new topics (antiangiogenic therapies resistance, MALT, GIST, RIRR, and ROS scavenging molecules), we also included new cited papers but anyway, the reviewer is welcome to directly suggest interesting papers that we may have missed.

Specific comments:

Abstract

Lines 18-20:  wrong definition of vasculogenic mimicry. The correct definition is given later on (lines 125-127).

The definition was corrected;

1 Introduction

Lines 31-33. The authors have forgotten non-angiogenic growth by vessel co-option!!

We are sorry for our forgetfulness; therefore, we added a more complete list of tumor vascularization mechanisms including the vessel co-option;

2 Tumour angiogenesis and vasculogenesis

Lines 86-88: This is by now incorrect. Hypoxia is present also in non-angiogenic tumours.

In this line we did not aim to describe when hypoxia is induced by malignancies but how the hypoxic microenvironment might induce the angiogenic response; however, as we did not mention the non-angiogenic tumors in our article, we added the word “often” to be more correct;

Lines 105-123: Several mechanisms of resistance to anti angiogenic treatment have been now described. The authors should quote and briefly discuss them

As suggested the mechanisms of resistance to the anti-angiogenic therapies were briefly described and discussed at the end of the paragraph;

3 Vasculogenic mimicry

Lines 128-129: Mosaic lining of the vasculogenic mimicry channels happens but not always!!

The sentence was accordingly modified;

Lines 153-156: It is correct that it is a mechanism of resistance to anti angiogenic treatment but not the only or  commonest (see comment above).

We added the word “often” to let the reader know that it is not the only anti-angiogenic resistance mechanism. Moreover, as previously reported, we added a new brief section on the anti-angiogenic resistance pathways;

Lines 156-195. This sections perhaps should be self-contained, as possible new approaches to gastric cancer. It begins with drugs which could possibly interfere with VM and than move on to other drugs, to end with anti angiogenic approaches. It should not be in the VM paragraph.

This section was accordingly moved into a new paragraph;

4 Epithelial to Endothelial transition

Lines 199-200: “Hypoxia is the best known and most potent VM inducer,” : reference 52 does not discuss vasculogenic mimcry.

The reference was accordingly changed;

Lines 215-216: This event can happen, but not always.

The sentence was changed accordingly;

5 ROS induced vascularization.

Line 229-231: Again, the non angiogenic growth by vessel co-option is omitted, making this sentence rather misleading.

The sentence was consequently removed;

  1. The Role of NADPH Oxidase in GC

Lines 358-359 “ Finally, ROS-induced angiogenesis is still a debated topic and therefore few drugs are currently investigated to inhibit such a phenomenon. “

This statement is puzzling as already in the abstract the authors stated:

“ In  such a particular condition, reactive oxygen species play a fundamental role, enhancing tumour proliferation, migration, inducing a glycolytic phenotype and promoting angiogenesis.”

The sentence was consequently removed;

Reviewer 2 Report

Thank you for inviting me to review the manuscript “The Role of Reactive Oxygen Species in Gastric Cancer Vascularization” submitted to the journal Biomolecules.

The authors prepared a well-written review on abnormal vascularization process in gastric cancer aiming to assess pathological processes leading to epithelial-to-endothelial transition, role of reactive oxygen species, and the effect of chemotherapies aimed to control tumor-induced vascularization.

I only have a couple of concerns about the manuscript:

1    1) Figure 1 does not add much value to the manuscript as it overly simplifies the relationships between the NOX and DUOX and their interactions HIF-1α and HIF-2α in GC cells. The role of ROS scavengers (superoxide dismutase and catalase) should be acknowledged and discussed thoroughly in the text and represented on a figure.

2)   Authors discussed the role of mitochondrial ROS as well as ROS produced by NOX enzymes, however they have not discussed the phenomenon termed ROS-induced ROS release that results in ROS amplification. Also, when discussing the effects of ROS, the authors do not specify whether they talk about endogenous or exogenous ROS.

Author Response

The authors prepared a well-written review on abnormal vascularization process in gastric cancer aiming to assess pathological processes leading to epithelial-to-endothelial transition, role of reactive oxygen species, and the effect of chemotherapies aimed to control tumor-induced vascularization.

I only have a couple of concerns about the manuscript:

1    1) Figure 1 does not add much value to the manuscript as it overly simplifies the relationships between the NOX and DUOX and their interactions HIF-1α and HIF-2α in GC cells. The role of ROS scavengers (superoxide dismutase and catalase) should be acknowledged and discussed thoroughly in the text and represented on a figure.

2)   Authors discussed the role of mitochondrial ROS as well as ROS produced by NOX enzymes, however they have not discussed the phenomenon termed ROS-induced ROS release that results in ROS amplification. Also, when discussing the effects of ROS, the authors do not specify whether they talk about endogenous or exogenous ROS.

We thank the reviewer for the precious comments and suggestions provided and for the time spent reading and improving our article. As requested we added a paragraph discussing the role of ROS scavengers (superoxide dismutase and catalase) which was consequently represented in the figure while ROS-induced ROS were discussed in deep in the paragraph “The Role of NADPH Oxidase in GC”. Generally, apart from the case of radiotherapy, all the sources of ROS included in the article are endogenous as all the described factors are inducers of intracellular ROS generation. However, as requested, we specified when needed in the text whether we discussed endogenous or exogenous ROS case by case.

Round 2

Reviewer 1 Report

I understand the problem of balancing two different opinion from reviewers!

It is a  much improved paper and he increase discussion of ROS and of different gastric neoplasms makes it now a quite original review.

Sorry one last tiny correction.

Lines 33-34:  Instead of  “….generating an intricate net of vessels composed of a mosaic of endothelial and tumours cells. “

write

“……generating an intricate net of vessels, some of which are composed of a mosaic of endothelial and tumour cells. “

As not all the vessels are mosaics

Well done!